# Prompt Pre-Training with Twenty-Thousand Classes for Open-Vocabulary Visual Recognition

**Shuhuai Ren**[‡†]**, Aston Zhang**[†]***, Yi Zhu**[†]**, Shuai Zhang**[†]**, Shuai Zheng**[†]**,**
**Mu Li**[†]**, Alex Smola**[†]**, Xu Sun**[‡]
[‡]National Key Laboratory for Multimedia Information Processing,
School of Computer Science, Peking University
[†]Amazon Web Services

## Abstract

This work proposes POMP, a prompt pre-training method for vision-language models. Being memory and computation efficient, POMP enables the learned prompt to condense semantic information for a rich set of visual concepts with over twenty-thousand classes. Once pre-trained, the prompt with a strong transferable ability can be directly plugged into a variety of visual recognition tasks including image classification, semantic segmentation, and object detection, to boost recognition performances in a zero-shot manner. Empirical evaluation shows that POMP achieves state-of-the-art performances on 21 datasets, e.g., $67.0\%$ average accuracy on 10 classification datasets ($+3.1\%$ compared to CoOp) and $84.4$ hIoU on open-vocabulary Pascal VOC segmentation ($+6.9$ compared to ZSSeg). Our code is available at `https://github.com/amazon-science/prompt-pretraining`.

## 1 Introduction

It has been a new norm to formulate visual recognition tasks (e.g., image classification, object detection, and semantic segmentation) as language-guided visual recognition or vision-and-language problems [42, 19, 61]. In language-guided visual recognition, categories of images are represented by natural language rather than discrete label IDs, and the semantics between images and their corresponding textual descriptions are often aligned via a contrastive loss during training [42]. Model inference also becomes an image-to-text matching problem, where text prompts like "`a photo of a [CLASSNAME]`" are curated as text descriptions of images. By varying the `[CLASSNAME]` placeholder and computing the similarity between text descriptions and images, we can identify the most suitable class name and consider it as the predicted target class. A significant benefit of this language-guided paradigm is that it supports open-vocabulary inference, that is, zero-shot recognition for arbitrary categories that may not even have been seen during training, thanks to the flexibility in modifying the class names in the textual prompt [42, 45].

As the context for class names, the text prompt plays a critical role in language-guided visual recognition models. A good prompt should holistically express the semantics of visual categories to better elicit the knowledge learned by vision-language models (VLMs) during pre-training. There are two popular types of prompts: hard prompts (e.g., `a photo of a [CLASSNAME]`) and soft prompts. Soft prompts are learnable token embeddings that can be fine-tuned given some input data, and have been demonstrated to be more effective and stable on downstream tasks than hard prompts [65, 35, 63]. However, traditional prompt tuning methods usually fine-tune the soft prompt on task-specific datasets with a limited number of class labels, making it difficult to generalize to novel classes and across tasks [45, 2]. For example, when transferred between the two downstream datasets of Flowers102 [40]

---

*Correspondence to Aston Zhang <az@astonzhang.com>

37th Conference on Neural Information Processing Systems (NeurIPS 2023).

and DTD [9], the soft prompt fine-tuned on DTD only achieves $33.4\%$ accuracy on Flowers102, significantly lower than the $61.8\%$ accuracy of a hand-crafted prompt on Flowers102, demonstrating a severe overfitting issue [49].

In this work, we aim to learn a universal soft prompt that covers a broad set of visual concepts while being task-agnostic. Specifically, we propose PrOMpt Pre-training (**POMP**), a method for scaling up prompt learning on the ImageNet-21K dataset, which has over twenty-thousand classes organized by the WordNet [37] hierarchy. The set of classes in ImageNet-21K includes general and long-tail visual categories of various semantic granularities, which have been proven to provide better downstream results for large models [30, 14]. Pre-training on this large-scale dataset helps condense semantic information into the soft prompt for universal visual discrimination. Once pre-trained, this universal prompt (i) can be easily applied to downstream datasets to improve model performance in zero-shot settings; (ii) is compatible with both region-level and pixel-level visual patterns, making it useful for various vision tasks such as object detection and semantic segmentation.

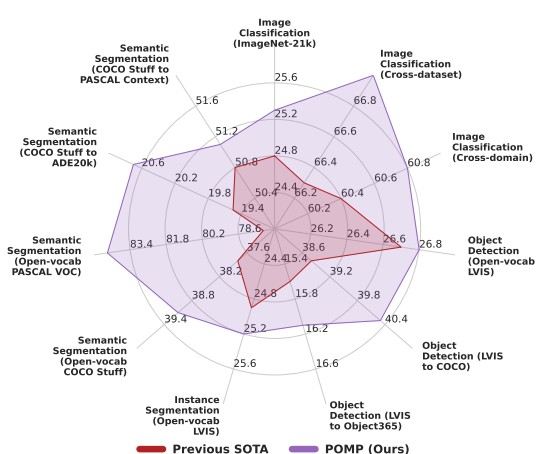

Figure 1: POMP outperforms previous state-of-the-art models on a broad range of visual recognition tasks and datasets.

However, pre-training prompt with such a massive class set is challenging due to generally prohibitive computational costs. During prompt pre-training, the activation of the whole text encoder needs to be kept independently for every class and the memory consumption increases proportionally to the number of classes. In short, pre-training prompts on ImageNet-21K requires over 300 GB GPU memory with traditional methods like CoOp [65]. In POMP, we solve this issue with a simple class sampling strategy, *local contrast*, which reduces the GPU memory requirement dramatically to less than 16 GB. Moreover, we propose a *local correction* strategy to reduce the bias caused by class sampling and improve the generalization of the pre-trained prompt.

Experimental results in Figure 1 show that POMP outperforms previous state-of-the-art (SOTA) models on a broad range of visual recognition tasks and datasets. Specifically, compared to zeroshot CLIP [42], POMP improves the accuracy on ImageNet-21K by a gain of $+2.9\%$. It also achieves an average accuracy of $67.0\%$ when transferred to 10 downstream image classification datasets, which is $3.1\%$ higher than CoOp [65]. For semantic segmentation, POMP achieves 39.1 hIoU on open-vocab COCO Stuff and 84.4 hIoU on open-vocab Pascal VOC, outperforming ZSSeg [61] by $+1.3$ and $+6.9$ hIoU, respectively. For object detection, POMP achieves 57.9 and 22.9 $AP_{50}$ when transferred from LVIS to COCO and Object365, surpassing Detic [67] by $+1.9$ and $+0.8$ $AP_{50}$, respectively.

## 2 Related Work

### 2.1 Language-Guided Visual Recognition

Language-guided visual recognition usually leverages VLMs as foundation models. Representative VLMs like CLIP [42] consist of an image encoder and a text encoder, which are used to encode image-text pairs into a joint feature space for learning the semantic alignment between vision and language [7, 46]. After being pre-trained on large-scale image-text pairs, CLIP-like models [28, 62, 33] are able to map images to their corresponding language descriptions, allowing visual recognition to generalize in the wild. This language-driven modeling paradigm also facilitates other vision tasks, including semantic segmentation [61, 32, 43, 13] and object detection [19, 15, 67]. These works typically designed a two-stage framework: it first leverages the pre-trained proposal network to extract features of specific visual patterns (e.g., segment mask and region) and then conducts classification in the same matching style as CLIP. The class descriptions for matching are synthesized using prompts, and in this work, we pre-train a soft prompt for VLMs on ImageNet-21K to further enhance their zero-shot generalization ability.

## 2.2 Prompt Tuning

In order to adapt VLMs to downstream tasks, recent research proposed a parameter-efficient tuning method named prompt tuning. CoOp [65] first proposed to replace the hand-crafted prompt with learnable vectors (also known as a soft prompt) for fine-tuning while freezing the entire pre-trained parameters. VPT [12], on the contrary, moved the learnable vectors from the text side to the image side, and proposed to concatenate the "visual soft prompt" and the patch sequence of an image as the input for fine-tuning. Prompt tuning was also used in other visual recognition tasks, such as object detection [15], semantic segmentation [43], and video recognition [39]. Over manual prompt engineering, the soft prompt optimized with few-shot data has achieved significant performance improvements, but only fitting one specific downstream dataset.

To enhance the generalization of the soft prompt to a wider range of unseen classes and datasets, CoCoOp [66] modeled the context condition on input images. A recent work MaPLe [29] appended the soft prompt to the hidden representations at each layer in both the text and image encoder. In sharp contrast to previous approaches, we propose to pre-train a universal soft prompt on large-scale datasets with massive visual categories. Such a pre-trained prompt is task-agnostic, allowing for direct transfer to various downstream datasets without fine-tuning.

## 3 Method

We first review the process of classical prompt tuning for VLMs in § 3.1. To address the training efficiency issue of previous methods, in § 3.2 we introduce our method of prompt pre-training (POMP) that includes two key components: local contrast and local correction. Our pre-trained prompt can then be transferred to downstream datasets and tasks in a zero-shot manner as discussed in § 3.3.

### 3.1 Preliminaries

Language-guided visual recognition models like CLIP formulate image classification as an image-text matching problem, where the goal is to select the correct textual class name from a predefined class set for the image query. Following [65, 29], we consider CLIP as our vision-language foundation model, for its simplicity in design and wide applicability. CLIP consists of an image encoder $f_I$ and a text encoder $f_T$. Given a visual recognition dataset $\mathcal{D}$ with a class set of $N$ class names $\mathcal{C} = \{c_i\}_{i=1}^{N}$, CLIP manually devises a hard prompt to synthesize textual descriptions $t_i$ for each class name $c_i$, e.g., $t_i =$ "a photo of a $[c_i]$". Then each class description is fed into the text encoder to generate the normalized class feature $\mathbf{w}_i = f_T(t_i)/\|f_T(t_i)\|_2 \in \mathbb{R}^d$, where $d$ is the dimension of the feature. The concatenation of $N$ class features $[\mathbf{w}_1, \cdots, \mathbf{w}_N] \in \mathbb{R}^{N \times d}$ can be considered as the class weight of a linear classifier for classifying an image. Given an input image $x$, the image encoder is used to extract its visual feature $\mathbf{x} = f_I(x)/\|f_I(x)\|_2 \in \mathbb{R}^d$. Finally, CLIP calculates the similarity between $\mathbf{x}$ and all the class features, then predicts the class with the highest similarity as the target class.

To address the inefficient expressiveness of the manual prompt, previous research like CoOp [65] proposed to parameterize the manual prompt as a soft prompt $\mathbf{\Theta}$, and fine-tune it to fit downstream datasets. The soft prompt is made up of a sequence of learnable token embeddings $\mathbf{\Theta} = [\boldsymbol{\theta}_1, \boldsymbol{\theta}_2, \cdots, \boldsymbol{\theta}_M] \in \mathbb{R}^{M \times e}$, where $M$ is a hyperparameter specifying the length of the soft prompt and $e$ is the dimension of the token embedding. The token embeddings of each class name $c_i$ are further appended to the soft prompt $\mathbf{\Theta}$ to generate the class feature $\mathbf{w}_i^{(\mathbf{\Theta})}$, and the prediction probability for the ground-truth class $y$ is denotes as

$$P(y \mid \mathbf{x}; \mathbf{\Theta}) = \frac{\exp(\mathbf{x}^\top \mathbf{w}_y^{(\mathbf{\Theta})}/\tau)}{\sum_{i=1}^{N} \exp(\mathbf{x}^\top \mathbf{w}_i^{(\mathbf{\Theta})}/\tau)}, \tag{1}$$

where $\mathbf{x}^\top \mathbf{w}_i$ represents the similarity score and $\tau$ is a temperature parameter. The parameters of the soft prompt are updated by minimizing the cross-entropy loss [65]:

$$\mathcal{L}(\mathbf{\Theta}) = \mathbb{E}_{(\mathbf{x},y)\in\mathcal{D}} \left[ -\log P(y \mid \mathbf{x}; \mathbf{\Theta}) \right]. \tag{2}$$

The gradient of $\mathcal{L}(\mathbf{\Theta})$ is represented as:

$$\nabla_{\mathbf{\Theta}} \left( -\log P(y \mid \mathbf{x}; \mathbf{\Theta}) \right) = \frac{1}{\tau} \left[ -\nabla_{\mathbf{\Theta}}(\mathbf{x}^\top \mathbf{w}_y^{(\mathbf{\Theta})}) + \sum_{i=1}^{N} P(y_i \mid \mathbf{x}; \mathbf{\Theta}) \nabla_{\mathbf{\Theta}}(\mathbf{x}^\top \mathbf{w}_i^{(\mathbf{\Theta})}) \right]. \tag{3}$$

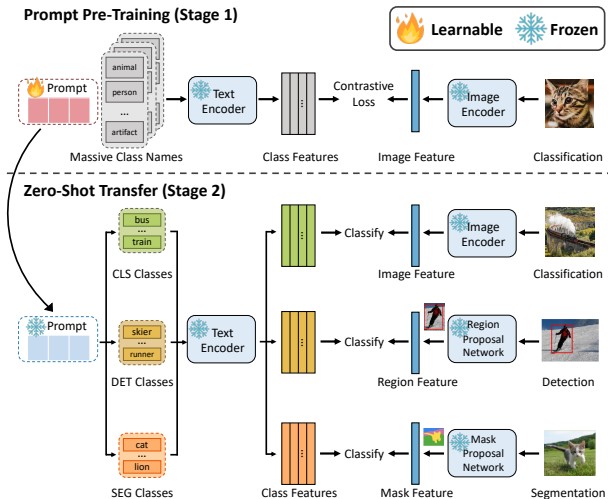

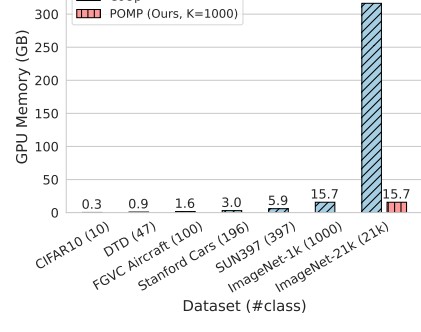

Figure 2: Overview of POMP. POMP pre-trains a soft prompt (🔥:learnable) on the ImgaNet-21K dataset with massive classes, and then directly transfers the learned prompt (❄:frozen) to downstream datasets of image classification (CLS), object detection (DET), and semantic segmentation (SEG) tasks. For DET and SEG, the region and mask proposal networks require pre-training with POMP prompt on detection and segmentation source data, respectively (See Appendix B).

Figure 3: GPU memory overhead (GB) required for prompt tuning on datasets with varying numbers of classes. The memory cost of CoOp on ImageNet-21K is 316.4 GB, which is generally prohibitive. POMP reduces the cost dramatically to 15.7 GB with *local contrast* among the 1000 sampled classes for optimization.

It can be decomposed into positive reinforcement for the ground-truth class and negative reinforcement for every class, which are the first and the second terms inside the square brackets of (3), respectively.

Note that both the image encoder and the text encoder are frozen during the prompt tuning process, which allows adapting the soft prompt efficiently to downstream data with very few learnable parameters. Methods along this direction of soft prompt learning include CoCoOp [66] and MaPLe [29]. However, most previous works fine-tune *task-specific* prompts, which limits their versatility and generalization [49].

## 3.2 POMP: Prompt Pre-Training

Now we present our task-agnostic prompt pre-training method: POMP. Once pre-trained, the learned prompt can be directly used for downstream tasks without fine-tuning (see Figure 2). As introduced in § 1, we propose to pre-train the soft prompt on the ImageNet-21K dataset for universal visual discrimination. Although the prompt tuning methods such as COOP and CoCoOp are parameter-efficient, they still incur computationally prohibitive training costs when applied to large-scale datasets with massive number of classes. Recall that the learnable parameters $\Theta$ are embedded in the text input, while the loss is calculated at the output layer of the text encoder. For every class description, we need to allocate nearly 15 MB of GPU memory to preserve the state of the entire frozen encoder (Transformer-base [53] with 12 layers), and propagate the gradient back through the last layer to the first layer, to update the soft prompt $\Theta$. Accordingly, the computational and caching cost of prompt tuning is proportional to the number of classes $N$. As shown in Figure 3, for general large-scale datasets like ImageNet-21K with more than twenty-thousand classes, traditional tuning methods will allocate 21K ×15 MB (more than 300 GB) of GPU memory, which is generally prohibitive.

To enable prompt tuning on massive classes and acquire the capability of global visual discrimination, we introduce a training-efficient algorithm called POMP, which reduces the GPU memory and training time of prompt tuning dramatically. POMP has two major components: **local contrast** and **local correction**. The former decreases the number of classes for contrastive learning through negative class sampling, and the latter reduces the bias caused by local contrast by adjusting the similarity scores of negative classes. We detail these two components in the following.

### 3.2.1 Local Contrast

Discriminating all classes during contrastive learning is the source of training inefficiency. In order to alleviate this problem, we propose to narrow the scope of contrastive learning from global to local, and only require the model to identify the ground-truth class of the input image from a subset of the full class set. The class subset is sampled at each training step, allowing the model to discriminate within an ever-changing set of categories, and gradually restoring the relationship among all categories.

Specifically, given an input image, we sample $K$ classes ($K$ is much smaller than the total number of classes, $N$), including the ground-truth class $y$ and $K-1$ negative classes. We use a straightforward yet effective proposal distribution of uniform distribution for negative class sampling, where every negative class has an equal probability, i.e., $p = 1/(N-1)$, of being sampled. We also explore alternative types of proposal distribution, such as frequency-based and similarity-based distributions. However, our experiments reveal that POMP with the simple uniform distribution considers both common and rare classes, as well as easy and difficult classes, resulting to the best performance. Please refer to Appendix E.1 for details.

After sampling, we denote the set of the negative classes as $\mathcal{N}$, with $|\mathcal{N}| = K - 1$. By using the local contrast, we can significantly reduce the training overhead to a fraction of $K/N$ of the original one. Upon completion of training, we can use the full class set to compute the prediction probability of each image. Overall, the motivation behind our local contrast is analogous to that in the NCE-based contrastive learning frameworks [52, 57]. In these frameworks, the contrast is performed by sampling a batch of instances due to computational limitations and computing the loss within the batch as an empirical estimation for the expected contrastive loss [25, 1, 22, 51, 68].

### 3.2.2 Local Correction

Given that the local contrast component necessitates a reduced number of negative classes, the negative reinforcement in the vanilla gradient in (3) is diminished to $K/N$. As a result, the prompt optimization direction is inevitably biased due to the absence of other negative classes. To mitigate this bias and enhance the model performance, we add a local correction term $m$ to the logits of the sampled negative classes $\mathbf{x}^\top \mathbf{w}_i^{(\boldsymbol{\Theta})}/\tau$ ($i \neq y$), which serves as a margin [60, 45, 69] between the positive and the negative logits. Accordingly, the final prediction probability of POMP is denoted as:

$$\tilde{P}\left(y \mid \mathbf{x}; \boldsymbol{\Theta}\right) = \frac{\exp(\mathbf{x}^\top \mathbf{w}_y^{(\boldsymbol{\Theta})}/\tau)}{\exp(\mathbf{x}^\top \mathbf{w}_y^{(\boldsymbol{\Theta})}/\tau) + \sum\limits_{i \sim \mathcal{N}} \exp(\mathbf{x}^\top \mathbf{w}_i^{(\boldsymbol{\Theta})}/\tau + m)}. \tag{4}$$

The local correction term $m$ encourages the positive logit to be larger than the negative logits by a certain margin, resulting in a more stringent decision boundary:

$$C_+ : \mathbf{x}^\top \mathbf{w}_y^{(\boldsymbol{\Theta})}/\tau \geq \mathbf{x}^\top \mathbf{w}_i^{(\boldsymbol{\Theta})}/\tau + m, \quad i \neq y.$$

Therefore, compared to the prediction probability without local correction, (4) makes the decision boundary more robust against the unsampled negative classes and enforces the learning of more discriminative class features [60]. This will improve model regularization and robustness across datasets and domains (to be shown in § 4.4). Different from other margin-based losses that use a fixed margin [11, 54], our margin $m$ is designed to adaptively adjust itself based on the value of $K$:

$$m = -\log\left((K-1)/(N-1)\right). \tag{5}$$

It is worth noting that $m$ in (5) is a positive scalar. When $K = N$, all classes are included during optimization, and $m$ equals zero. In this case, (4) degenerates to the standard prediction probability in (1). As the value of $K$ decreases and the number of visible classes is reduced, the margin $m$ increases to create space for potential class features in the representation space. Our adaptive margin outperforms the fixed margins (which are specified as hyper-parameters) for various $K$, allowing models to maintain optimal performance under different computing budgets (to be shown in § 4.4).

### 3.3 Zero-Shot Transfer Learning

As shown in Figure 2, after pre-training, our POMP prompt can be used to synthesize class features for classification with an arbitrary class set, supporting zero-shot inference on downstream datasets

and tasks. In order to plug the POMP prompt into other visual tasks like semantic segmentation and object detection, we adopt a **two-stage framework**. In stage one, we use a pre-trained proposal network to generate a set of mask or region proposals. In stage two, we classify each proposal with the class features generated by our POMP prompt. Experiments in § 4.3 will show that the POMP prompt can handle both pixel-level and region-level visual patterns, leading to improved performance in segmentation and detection tasks.

# 4 Experiments

## 4.1 POMP Prompt Pre-Training

We take CLIP (ViT/B-16) [42] as the backbone and conduct prompt pre-training on the ImageNet-21K dataset. The number of training samples for each class is 16 (16 shots), and the prompt length is 16. We sample 1,000 classes at each training step, i.e., $K = 1000$ in (4). See Appendix A for details.

## 4.2 End Task Setups and Implementation Details

To evaluate the generalization of the pre-trained prompt, we directly transfer it to downstream tasks and datasets. Appendix C lists the details of all the datasets. We follow previous works [58, 61, 19, 67] to designate data belonging to two class sets as **source data** and **target data**, respectively. The proposal networks are pre-trained on the source data with the source class set, while conducting zero-shot evaluation on the target data with the target class set. There are two protocols for the source-target data split. The first is the **open-vocabulary protocol**, where the class set of one dataset is divided into two disjoint groups for the source and target data. The second protocol is the **cross-dataset protocol**, in which the source and target data are from two independent datasets with potentially overlapping class sets. See Appendix B for implementation details.

## 4.3 Results and Analysis

### 4.3.1 Prompt Pre-Training on ImageNet-21K

Table 1 shows the results of POMP prompt on the ImageNet-21K test set after pre-training. The traditional prompt learning methods (e.g., CoOp and MaPLe) are trained on the ImageNet-1K dataset due to their prohibitive computational cost if trained on ImageNet-21K (more than 300 GB of GPU memory). On the contrary, our POMP prompt, pre-trained on ImageNet-21K using less than 16 GB of GPU memory, achieves the highest accuracy of $25.3\%$ based on the CLIP (ViT-B/16) backbone, which surpasses Zeroshot-CLIP by $3.5\%$ and Linear Probe by $4.4\%$. The VPT method, which uses visual prompts on the image side, does not require training overhead proportional to the number of classes, making it applicable to the ImageNet-21K dataset. VPT prepends independent learnable vectors to the hidden states of each layer in the visual backbone, surpassing linear probing and the previous

Table 1: Performance on the ImageNet-21K test set. ZeroshotCLIP and Prompt Ensemble in the top block conduct zero-shot inference. CoOp and MaPLe, indicated in gray in the middle block, are trained on the ImageNet-1K dataset due to prohibitive GPU memory consumption if trained on ImageNet-21K. The remaining methods in the bottom block are trained on ImageNet-21K.

| Method | ResNet50 | ViT-B/32 | ViT-B/16 |
|---|---|---|---|
| ZeroshotCLIP [42] | 17.5 | 19.8 | 21.8 |
| Prompt Ensemble [42] | 18.8 | 20.9 | 23.5 |
| CoOp [65] | 16.6 | 18.1 | 20.8 |
| MaPLe [29] | - | 21.6 | 24.2 |
| Linear Probing [42] | 6.5 | 18.2 | 20.9 |
| VPT [12] | - | 21.8 | 24.8 |
| POMP (Ours) | **20.2** | **22.2** | **25.3** |

prompt tuning methods. However, its performance based on ViT-B/16 is still $0.5\%$ worse than ours, demonstrating that our POMP prompt can better distinguish a large number of general visual categories. In addition, our method is agnostic to the backbone architectures like ResNet and ViT, and the improvement is consistent.

**Cross-dataset and Cross-domain Image Classification.** Our POMP prompt, which has been pre-trained on a large number of classes, demonstrates a strong generalization ability. As shown in Table 2, POMP achieves the highest average accuracy of $67.0\%$ when transferred to 10 downstream image classification datasets, outperforming CoOp by $3.1\%$ and surpassing the previous SOTA in

Table 2: Cross-dataset and cross-domain evaluation for image classification. The backbone is ViT/B-16. Overall, POMP achieves the highest average accuracy, indicating better generalization.

| | Target (cross-dataset) | | | | | | | | | | | Target (cross-domain) | | | | |
| --- | --- | --- | --- | --- | --- | --- | --- | --- | --- | --- | --- | --- | --- | --- | --- | --- |
| | Caltech101 | OxfordPets | StanfordCars | Flowers102 | Food101 | Aircraft | SUN397 | DTD | EuroSAT | UCF101 | Average | ImageNetV2 | ImageNet-S | ImageNet-A | ImageNet-R | Average |
| hard prompt | 93.3 | 88.2 | 65.6 | 67.4 | 85.3 | 23.7 | 62.6 | 44.3 | 42.0 | 65.1 | 63.7 | 60.9 | 46.1 | 47.8 | 74.0 | 57.2 |
| CoOp [65] | 93.7 | 89.1 | 64.5 | 68.7 | 85.3 | 18.5 | 64.2 | 41.9 | 46.4 | 66.6 | 63.9 | 64.2 | 48.0 | 49.7 | 75.2 | 59.3 |
| CoCoOp [66] | 94.4 | 90.1 | 65.3 | 71.9 | 86.1 | 22.9 | 67.4 | 45.7 | 45.4 | 68.2 | 65.7 | 64.1 | 48.8 | 50.6 | 76.2 | 59.9 |
| LASP [5] | 94.5 | 89.4 | 64.8 | 70.5 | 86.3 | 23.0 | 67.0 | 45.5 | 48.3 | 68.2 | 65.8 | 63.8 | 49.0 | 50.7 | 77.1 | 60.1 |
| VPT [12] | 93.7 | **90.6** | 65.0 | 70.9 | 86.3 | 24.9 | 67.5 | 46.1 | 45.9 | 68.7 | 66.0 | **64.2** | 49.2 | 51.3 | 77.0 | 60.4 |
| MaPLe [29] | 93.5 | 90.5 | 65.6 | 72.2 | 86.2 | 24.7 | 67.0 | **46.5** | 48.1 | **68.7** | 66.3 | 64.1 | 49.2 | 50.9 | 77.0 | 60.3 |
| POMP (Ours) | **95.0** | 89.5 | **66.8** | **72.4** | **86.3** | **25.6** | **67.7** | 46.2 | **52.1** | 68.5 | **67.0** | 63.8 | **49.8** | **51.6** | **77.9** | **60.8** |

7/10 datasets. This is due to the fact that, after learning with enormous long-tail categories, POMP can provide a more expressive context for fine-grained visual concepts such as specific *objects* and *scenes*, resulting in improved performance on datasets like StanfordCars ($+1.2\%$) and Aircraft ($+0.9\%$), as well as SUN397 ($+0.7\%$) and EuroSAT ($+4\%$). Furthermore, POMP is more robust to domain shift and achieves a new SOTA with $60.8\%$ accuracy on 4 out-of-domain variants of the ImageNet dataset.

**Training Efficiency.** POMP also achieves comparable accuracy to the classical prompt tuning methods when fine-tuning on specific downstream datasets, but significantly reduces the training cost. Table 3 shows the performance of prompt tuning on ImageNet-1K, using a visual backbone of ViT-B/16 and 16 shots. The epoch is 50 for CoOp and POMP, and 10 for CoCoOp. CoOp generates all the 1000 class features at each training step, which consumes 28 GB of memory and takes 5.9 hours to finish the fine-tuning. The training time of CoCoOp is

Table 3: Prompt tuning on ImageNet-1K. POMP ($K = 128$) achieves comparable accuracy with CoOp and CoCoOp, but using less than 19% GPU memory and 50% training time.

| Method | Acc. (%) | GPU Mem. (GB) | Training Time (h) |
| --- | --- | --- | --- |
| CoOp | 71.9 | 28.2 | 5.9 |
| CoCoOp | 70.1 | 28.3 | 27.5 |
| POMP ($K = 128$) | 71.2 | 5.3 | 2.7 |
| POMP ($K = 256$) | 71.4 | 8.8 | 3.3 |
| POMP ($K = 512$) | 71.6 | 15.9 | 4.2 |

even longer because it devises instance-specific prompts that require an independent forward pass for each image. Compared to these baselines, POMP ($K = 128$) achieves competitive accuracy on ImageNet-1K while using less than 19% of GPU memory and 50% of training time, demonstrating its superiority.

#### 4.3.2 Open-Vocabulary Semantic Segmentation

Table 4 shows the results of our method on open-vocabulary COCO Stuff and Pascal VOC. POMP outperforms the previous state-of-the-art method, ZSSeg [61], with a higher hIoU of 39.1 and mIoU-unseen of 38.2 on COCO Stuff. On Pascal VOC, the improvement of POMP is more significant with $+6.9$ hIoU and $+4.3$ mIoU-unseen. Figure 4 illustrates qualitative results on open-vocabulary COCO-Stuff, where POMP demonstrates a stronger ability to distinguish background categories compared to ZSSeg. For example, in case (1), ZSSeg misclassifies the classes of *playingfield* as *dirt*, while the POMP prompt with richer contextual semantics better expresses the difference between regular land and the playingfield with specific textures, thus facilitating the matching of the visual region with the ground-truth class.

POMP also demonstrates its generalization ability in cross-dataset settings. Taking standard COCO Stuff as the source dataset for mask proposal network pre-training, POMP achieves 20.7 mIoU and 51.1 mIoU when transferred to the target datasets of ADE20K and PASCAL Context, respectively, outperforming ZSSeg by $+1.3$ mIoU and $+0.3$ mIoU. Overall, POMP obtains remarkable gains over previous works in all settings.

#### 4.3.3 Open-Vocabulary Object Detection

Table 4: Comparison with state-of-the-art methods on COCO Stuff dataset and Pascal VOC dataset. POMP and ZSSeg share the same mask proposal network and training strategy.

| Method | Open-Vocab COCO Stuff | | | Open-Vocab Pascal VOC | | |
|---|---|---|---|---|---|---|
| | hIoU | mIoU | | hIoU | mIoU | |
| | | seen | unseen | | seen | unseen |
| SPNet [58] | 16.8 | 20.5 | 14.3 | 21.8 | 73.3 | 15.0 |
| ZS3 [4] | 15.0 | 34.7 | 9.5 | 28.7 | 77.3 | 17.7 |
| CaGNet [20] | 18.2 | 35.5 | 12.2 | 39.7 | 78.4 | 25.6 |
| ZegFormer [13] | 34.8 | 36.6 | 33.2 | 73.3 | 86.4 | 63.6 |
| ZSSeg [61] | 37.8 | 39.3 | 36.3 | 77.5 | 83.5 | 72.5 |
| POMP (Ours) | **39.1** | **39.9** | **38.2** | **84.4** | **93.6** | **76.8** |

Table 5: Cross-dataset evaluation for semantic segmentation. The mask proposal network is pre-trained on standard COCO Stuff.

| Method | Source Dataset: Standard COCO Stuff | | | Target Dataset: ADE20K | | | Target Dataset: PASCAL Context | | |
|---|---|---|---|---|---|---|---|---|---|
| | mIoU | fwIoU | pACC | mIoU | fwIoU | pACC | mIoU | fwIoU | pACC |
| ZSSeg [61] | 40.8 | 49.0 | 62.7 | 19.5 | 48.7 | 60.0 | 50.8 | 64.1 | 75.7 |
| POMP (Ours) | **41.1** | **49.2** | **62.9** | **20.7** | **51.5** | **63.7** | **51.1** | **65.4** | **76.1** |

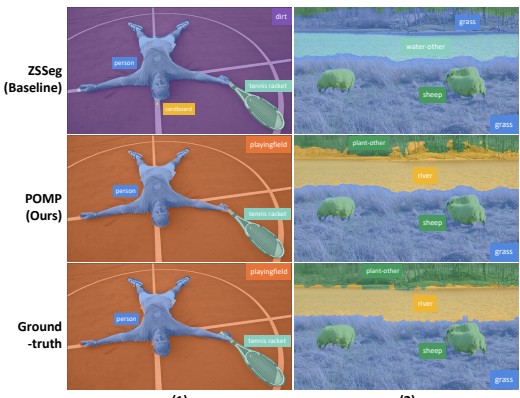

Figure 4: Qualitative results on open-vocabulary COCO-Stuff. Compared to ZSSeg, POMP correctly identifies the background category of `playingfield` (left) and `river` (right).

Table 6: Cross-dataset evaluation for object detection. The region proposal network is pre-trained on standard LVIS. POMP and Detic share the same region proposal network and training strategy.

| Method | Source Dataset: Standard LVIS | | | | | | Target Dataset: COCO | | | | | | Target Dataset: Objects365 | | | | | |
|---|---|---|---|---|---|---|---|---|---|---|---|---|---|---|---|---|---|---|
| | AP | $AP_{50}$ | $AP_{75}$ | $AP_s$ | $AP_m$ | $AP_l$ | AP | $AP_{50}$ | $AP_{75}$ | $AP_s$ | $AP_m$ | $AP_l$ | AP | $AP_{50}$ | $AP_{75}$ | $AP_s$ | $AP_m$ | $AP_l$ |
| ViLD* [19] | 27.5 | 41.8 | 29.3 | 20.6 | 35.9 | 43.4 | 34.1 | 52.3 | 36.5 | 21.6 | 38.9 | 46.1 | 11.5 | 17.8 | 12.3 | 4.2 | 11.1 | 17.8 |
| DetPro [15] | 28.4 | 42.9 | 30.3 | 21.0 | 36.7 | 44.1 | 34.9 | 53.8 | 37.4 | 22.5 | 39.6 | 46.3 | 12.1 | 18.8 | 12.9 | 4.5 | 11.5 | 18.6 |
| Detic [67] | 36.8 | 50.7 | 38.6 | 26.1 | 46.7 | 51.7 | 38.8 | 56.0 | 41.9 | 25.6 | 42.2 | 50.0 | 15.6 | 22.1 | 16.8 | 6.1 | 15.6 | 23.8 |
| POMP (Ours) | **37.2** | **51.1** | **39.3** | **26.5** | **47.2** | **52.6** | **40.3** | **57.9** | **43.6** | **28.3** | **43.9** | **50.6** | **16.1** | **22.9** | **17.3** | **6.2** | **16.3** | **24.7** |

We compare POMP with state-of-the-art methods on the open-vocabulary LVIS benchmarks and report results in Table 7. POMP achieves $AP_r$ of 26.8 for object detection and 25.2 for instance segmentation. See Appendix D for qualitative results. Under the cross-dataset setting, we pre-train the visual backbone on the source dataset of standard LVIS, and evaluate the recognition ability on COCO and Object365. As shown in Table 6, compared to Detic, POMP provides a gain of 1.9 $AP_{50}$ on COCO and 0.8 $AP_{50}$ on Object365, respectively.

Table 7: Comparison with previous SOTA on LVIS dataset. $AP_r$ is the main evaluation metric for open-vocabulary object detection.

| Method | Detection | | | | Instance segmentation | | | |
|---|---|---|---|---|---|---|---|---|
| | $AP_r$ | $AP_c$ | $AP_f$ | AP | $AP_r$ | $AP_c$ | $AP_f$ | AP |
| ViLD [19] | 16.7 | 26.5 | 34.2 | 27.8 | 16.6 | 24.6 | 30.3 | 25.5 |
| DetPro [15] | 20.8 | 27.8 | 32.4 | 28.4 | 19.8 | 25.6 | 28.9 | 25.9 |
| PromptDet [18] | - | - | - | - | 21.4 | 23.3 | 29.3 | 25.3 |
| Detic [67] | 26.7 | 36.4 | 40.3 | 36.3 | 24.9 | 32.5 | 35.6 | 32.4 |
| POMP (Ours) | **26.8** | 36.4 | 40.4 | 36.2 | **25.2** | 33.0 | 35.6 | 32.7 |

## 4.4 Ablation Study

We decouple the two components of local contrast and local correction in POMP, and conduct an ablation study to examine their individual contributions. Since removing the local contrast component will lead to prohibitive training cost, we investigate the impact of this component by varying the number of sampled classes $K$. As shown in Table 8, the performance of POMP improves as $K$ increases. As discussed in § 4.3.1 and Table 3, the local contrast component balances accuracy and cost by adjusting $K$.

Table 8: Ablation on the local contrast and local correction in POMP based on the CLIP (ViT/B-16).

| Method | ImageNet-21K | Cross-dataset (10 Avg.) | Cross-domain (4 Avg.) |
|---|---|---|---|
| POMP ($K = 100$) | 24.1 | 65.5 | 59.5 |
| POMP ($K = 500$) | 24.9 | 66.5 | 60.0 |
| POMP ($K = 1000$) | 25.3 | 67.0 | 60.8 |
| - local correction | 25.0 **(-0.3)** | 65.8 **(-1.2)** | 59.8 **(-1.0)** |

On the other hand, as shown in Table 8, removing the local correction component from POMP ($K = 1000$) results in a decline of 1.2 and 1.0 in the average accuracy of cross-dataset and cross-domain transfer, respectively. This indicates that local correction significantly improves the

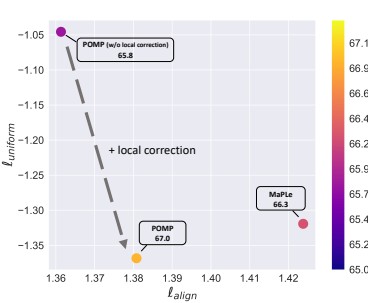

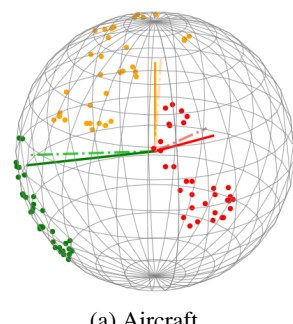

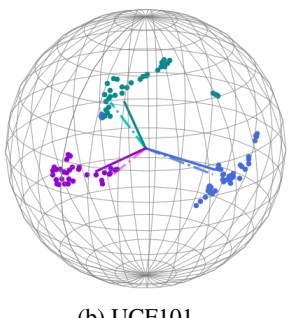

(a) Aircraft.        (b) UCF101.

Figure 5: $\ell_{\text{align}}$ and $\ell_{\text{uniform}}$ of POMP. For both measures, lower numbers are better. The color of circles and the numbers in the boxes denote the average cross-dataset accuracy over 10 datasets (higher is better).

Figure 6: Projection of image features (points), class features of POMP (intersections of solid lines and sphere) and class features of CoOp (intersections of light dash-dot lines and sphere). Each color represents a class. Class features of POMP have better alignment with centroids of the corresponding images, and are distributed with better uniformity.

generalization of the pre-trained prompt. Furthermore, we analyze the impact of the adaptive margin (5) in the local correction. We pre-train prompts on ImageNet-21K with varying $m$ values $(0, 0.5, 1, 1.5)$ and report the cross-dataset accuracy. Notably, our local correction method dynamically sets $m$ to 1.5 when $K = 319$, and $m$ to 1 when $K = 1000$. The results are shown in Table 9.

When the number of sampled classes is relatively small ($K = 319$), increasing the margin creates more space for potential negative classes, thereby improves cross-dataset accuracy. Conversely, for large $K$ values (e.g., 1000), imposing a very large margin ($m = 1.5$) disrupts the natural class distribution and diminishes generalization ability. Overall, compared to the fixed margins [11, 54], our adaptive margin decreases as $K$ increases, achieving optimal performance across different computing budgets (controlled by $K$) and sparing the time for extensive hyper-parameter search.

Table 9: Ablation on the adaptive margin $m$ in the local correction.

| m | K=319 | K=1000 |
|---|---|---|
| 0 | 65.2 | 65.8 |
| 0.5 | 65.7 | 66.1 |
| 1 | 66.2 | **67.0 (Ours)** |
| 1.5 | **66.5 (Ours)** | 66.4 |

See Appendix E for more ablation studies on the number of shots and prompt length.

## 4.5 Understanding the Pre-trained Prompt

To better understand the pre-trained prompt, we analyze the feature space of POMP through the properties of alignment and uniformity [56]. Intuitively, the image feature and its ground-truth class feature are supposed to stay closed (alignment). Besides, all the class features should be uniformly distributed to preserve maximal information and make the categories more distinguishable (uniformity). We use the alignment and uniformity loss in the vision-and-language field [45, 63] for representation probing. The alignment loss calculates the expected distance between features of an image $\mathbf{x}$ and its ground truth class $\mathbf{w}_y^{(\boldsymbol{\Theta})}$:

$$\ell_{\text{align}} \triangleq \mathop{\mathbb{E}}_{(\mathbf{x},y)\in\mathcal{D}} \left\| \mathbf{x} - \mathbf{w}_y^{(\boldsymbol{\Theta})} \right\|^2, \tag{6}$$

while the uniformity loss measures how well the class features $\mathbf{w}^{(\boldsymbol{\Theta})}$ are uniformly distributed:

$$\ell_{\text{uniform}} \triangleq \log \mathop{\mathbb{E}}_{\substack{1\leqslant i,j\leqslant N, \\ i\neq j}} \exp(-2\|\mathbf{w}_i^{(\boldsymbol{\Theta})} - \mathbf{w}_j^{(\boldsymbol{\Theta})}\|^2). \tag{7}$$

We visualize the alignment and uniformity measures of POMP and the previous SOTA, MaPLe, in Figure 5. For both measures, lower numbers are better. The circle of POMP in the figure is located in the lower left with the lightest color, indicating relatively smaller losses and the best performance under the cross-dataset setting. Compared with the method without local correction, POMP significantly reduces the uniformity loss at only a slight expense of alignment. In other words, our pre-trained prompt not only ensures the alignment of the image and the ground-truth class, but

also disperses the class features in the representation space, thereby improving the generalization and robustness of the model. The visualization of the feature space in Figure 6 also verifies our superiority. The endpoints of the POMP class features are closer to the centroids of the image features, indicating better alignment and reduced $\ell_{\mathrm{align}}$ loss (from 1.39 to 1.36 on Aircraft and from 1.41 to 1.36 on UCF101). Furthermore, the larger angles between the POMP class features demonstrate better feature uniformity and reduced $\ell_{\mathrm{uniform}}$ loss (from $-0.66$ to $-0.81$ on Aircraft and from $-0.95$ to $-1.23$ on UCF101) compared to CoOp.

## 5 Conclusion

We present POMP to pre-train a general soft prompt on ImageNet-21K for universal visual discrimination. The learned prompt can be easily plugged into various visual recognition datasets and tasks for zero-shot inference. Experiments on open-vocabulary image classification, semantic segmentation, and object detection show that POMP surpasses previous methods by a considerable margin.

## Limitations

To facilitate future research, we analyze the limitations in our work and propose potential solutions. **(1)** We present the local contrast and use the loss within a subsampled class set as an empirical estimation for the expected contrastive loss within the full class set. However, the theoretical risk of such an estimation is urged to be investigated. **(2)** ImageNet-21K comprises a vast number of classes that are organized based on a semantic structure. By leveraging the hyponym and hypernym relations provided by WordNet synsets, we can derive the parent class and a list of child classes for each class. We believe that utilizing the semantic information holds the potential to further enhance performance. **(3)** Despite the excellent performance exhibited by our pre-trained prompt, its interpretability poses a significant challenge because the context vectors are optimized in a continuous space. We leave it as future work.

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

# A  Pre-training Details

We conduct prompt pre-training on the ImageNet-21K dataset (official winter 2021 released version[2]). We follow the processing methods in [47], which involves cleaning invalid classes, allocating 50 images per class for a validation split, and crop-resizing all the images to 224 resolution. We conduct all the experiments on 8×Nvidia V100 GPUs. For pre-training, the learnable vector is randomly initialized by drawing from a zero-mean Gaussian distribution with a standard deviation equal to 0.02. We use the SGD optimizer with an initial learning rate of 0.002, decayed by the cosine annealing rule. The batch size is 32, and the maximum epoch is 20.

For the mask proposal network and region proposal network pre-training, we strictly follow the settings of ZSSeg [61] and Detic [67], respectively. Specifically, we take MaskFormer [8] with ResNet-101 [23] as the mask proposal network. We use an AdamW optimizer with the initial learning rate of $1e$-4, weight decay of 1e-4, a backbone multiplier of 0.1, and a poly learning rate policy with a power of 0.9. Besides, we take CenterNet2 [68] detector with ImageNet-21k pre-trained ResNet-50 [47] as the region proposal network. We use an Adam optimizer with learning rate $2e$-4. Other tricks like Federated Loss, repeat factor sampling, and large scale jittering are incorporated to further improve the performance. As with Detic, we leverage both region-level and image-level supervision. We always first train a converged base-class-only model ($4\times$ schedule) and fine-tune it with additional image-labeled data for another $4\times$ schedule.

# B  Setting for Segmentation and Detection

Table 10 outlines the settings for semantic segmentation and object detection. We further introduce the settings in detail from three perspectives: backbone, data processing, and prompt.

**Backbone.**  In general, we adopt a **two-stage framework** for these two tasks. At stage one, we use a pre-trained proposal network to generate a set of mask or region proposals. At stage two, we classify each proposal with the class features generated by our POMP prompt. For semantic segmentation, our POMP shares the same visual backbone as ZSSeg [61], which uses a pre-trained MaskFormer [8] with ResNet-101 [23] as default backbone to extract a set of binary masks. For object detection, our POMP shares the same visual backbone with Detic [67], which takes CenterNet2 [68] detector with ResNet-50 as its backbone, and leverages both region-level and image-level supervision.

**Data Processing.**  We follow previous work [58, 61, 19, 67] to designate data belonging to two class sets as **source data** and **target data**, respectively. The proposal networks are pre-trained on the source data with the source class set, while conducting zero-shot evaluation on the target data with the target class set. There are two protocols for the source-target data split. The first is the **open-vocabulary protocol**, where the class set of one dataset is divided into two disjoint groups for the source and target data, respectively. The second protocol is the **cross-dataset protocol**, in which the source and target data are from two independent datasets with potentially overlapping class sets.

We introduce the details of class set splitting in the open-vocabulary protocol. COCO Stuff and Pascal VOC 2012 are the two semantic segmentation datasets using the open-vocabulary protocol. Following previous settings [58, 61], a total of 171 annotated classes in COCO Stuff are divided into 156 seen classes and 15 unseen classes. For Pascal VOC 2012, a total of 20 classes are divided into 15 seen classes and 5 unseen classes, and the provided augmented annotations are used. LVIS is the object detection dataset using the open-vocabulary protocol. The standard LVIS dataset contains object detection and instance segmentation labels for 1203 classes. The classes are divided into three groups: frequent, common, and rare, based on the number of training images. According to previous work [19], the data from the 866 frequent and common classes are considered the source data, while those from the remaining 337 rare classes are the target data in testing. We note that Detic utilizes both box-supervised data from LVIS as well as image-supervised data from ImageNet-21K that overlaps with LVIS (997 classes, 277 of which are novel classes). This allows Detic to demonstrate transfer not only from base to novel classes, but also from image-level to box-level recognition. Since Detic is the closest existing method to ours that leverages ImageNet-21K, we chose it as a strong baseline and followed its setup for fair comparison.

---

[2]https://image-net.org/

Table 10: Settings for semantic segmentation and object detection.

| Task | Proposal Network | Setting | Source Data and Class Set (for proposal network pre-training) | Target Data and Class Set (for zero-shot evaluation) |
|---|---|---|---|---|
| Semantic Segmentation | MaskFormer (Mask Proposal Network) | Open-vocab COCO Stuff | COCO Stuff (seen) | COCO Stuff (unseen) |
| | | Open-vocab PASCAL VOC | PASCAL VOC (seen) | PASCAL VOC (unseen) |
| | | Cross-dataset | COCO Stuff | ADE20K / PASCAL Context |
| Object Detection | CenterNet2 detector (Region Proposal Network) | Open-vocab LVIS | LVIS (frequent+common) + ImageNet-21K (overlaps with LVIS) | LVIS (rare) |
| | | Cross-dataset | LVIS + ImageNet-21K (overlaps with LVIS) | COCO / Object365 |

**Prompt.** ZSSeg provides two kinds of prompts: hand-crafted prompts and learning-based prompts. Hand-crafted prompts include *single prompt*, i.e., "`a sculpture of a [CLASSNAME]`", as well as *ImageNet prompts* [42] and *ViLD prompts* [19], which are used for prompt ensemble and consist of $80$ and $14$ hard prompts, respectively. The learning-based prompt is obtained by fine-tuning a randomly initialized soft prompt on the source data. Accordingly, for a fair comparison, we conducted two sets of experiments based on whether to use the source data for prompt fine-tuning. (1) The results of ZSSeg with various hard-crafted prompts and the pre-trained POMP prompt without access to the source data can be found in Table 13 in Appendix E.3. (2) The results of ZSSeg with learning-based prompts initialized from random vectors and our pre-trained POMP prompt, both using source data for further fine-tuning, can be found in Table 4 and Table 5 in § 4.3.2. Detic has also extensively delved into intricate prompts, such as "`a photo of a [CLASS] in the scene`". Moreover, it has made endeavors to employ synonyms for each category. Nevertheless, its ultimate recommendation is to use a simple yet effective prompt, i.e., "`a [CLASSNAME]`", and all its released checkpoints are based on this prompt. We strictly adhere to Detic's best practice, the evaluation of Detic and POMP in § 4.3.3 are both conducted without any further prompt tuning on the source data.

## C  Datasets

The details of the downstream datasets for image classification, semantic segmentation, and object detection are shown in Table 11.

**Image Classification.** For cross-dataset image classification, we evaluate the performance of POMP on 10 downstream datasets, including Caltech-101 [17], Oxford-Pets [41], Stanford Cars [31], Oxford-Flowers102 [40], Food-101 [3], FGVC Aircraft [36], EuroSAT [24], SUN-397 [59], Describable Textures (DTD) [9], UCF-101 [50]. We also conduct zero-shot evaluation on 4 out-of-domain datasets derived from ImageNet [10], including ImageNetV2 [44], ImageNet-S [55], ImageNet-A [27], and ImageNet-R [26], to evaluate the domain generalization capability of our method.

**Semantic Segmentation.** We perform open-vocab semantic segmentation on COCO Stuff [6] and Pascal VOC 2012 [16]. Following previous notation and settings [58, 61], we split the class set into seen and unseen classes, where data for seen classes is considered the source data and data for unseen classes is considered the target data. The major measures for evaluation include mIoU and the harmonic mean IoU (hIoU) among both seen and unseen classes [61]. The hIoU is defined as:

$$\text{hIoU} = \frac{2 \times \text{mIoU}_{\text{seen}} \times \text{mIoU}_{\text{unseen}}}{\text{mIoU}_{\text{seen}} + \text{mIoU}_{\text{unseen}}}$$

We also conduct cross-dataset evaluation, which takes the standard COCO Stuff dataset as the source dataset for pre-training a mask proposal network, and then conducts zero-shot inference on ADE20K [64] and PASCAL Context [38].

**Object Detection.** We evaluate the performance of POMP on the object detection dataset LVIS [21] under the open-vocabulary setting proposed by [19]. The source data consists of box-level data from LVIS's 866 frequent and common classes, as well as image-level data from ImageNet-21K that overlaps with LVIS. The target data for testing comprises the remaining 337 rare classes in LVIS. We take $\text{AP}_r$, i.e., AP on rare classes, as the major measure. $\text{AP}_f$ and $\text{AP}_c$, i.e., AP on frequent and common classes, are also reported. In the cross-dataset setting, the region proposal network is

Table 11: Datasets in our experiments.

| Dataset | Classes | Train Size | Test Size | Metric |
|---|---|---|---|---|
| *Datasets of Image Classification* | | | | |
| Caltech-101 [17] | 102 | 3,060 | 6,086 | mean per-class accuracy |
| Oxford-IIIT Pets [41] | 37 | 3,680 | 3,669 | mean per-class accuracy |
| Stanford Cars [31] | 196 | 8,144 | 8,041 | accuracy |
| Oxford Flowers-102 [40] | 102 | 2,040 | 6,149 | mean per-class accuracy |
| Food-101 [3] | 101 | 75,750 | 25,250 | accuracy |
| FGVC Aircraft [36] | 100 | 6,667 | 3,333 | mean per-class accuracy |
| SUN-397 [59] | 397 | 15,880 | 19,850 | accuracy |
| Describable Textures (DTD) [9] | 47 | 3,760 | 1,880 | accuracy |
| EuroSAT [24] | 10 | 10,000 | 5,000 | accuracy |
| UCF-101 [50] | 101 | 7,639 | 3,783 | accuracy |
| ImageNetV2 [44] | 1,000 | 10,000 | 10,000 | accuracy |
| ImageNet-S [55] | 1,000 | 50,889 | 50,889 | accuracy |
| ImageNet-A [27] | 200 | 7,500 | 7,500 | accuracy |
| ImageNet-R [26] | 200 | 30,000 | 30,000 | accuracy |
| *Datasets of Semantic Segmentation* | | | | |
| COCO Stuff [6] | 171 | 117K | 5K | mIoU (seen/unseen), hIoU |
| PASCAL VOC [16] | 20 | 11,185 | 1,449 | mIoU (seen/unseen), hIoU |
| ADE20K [64] | 150 | 20K | 3K | mIoU, fwIoU, pACC |
| PASCAL Context [38] | 59 | 10,103 | 9,637 | mIoU, fwIoU, pACC |
| *Datasets of Object Detection* | | | | |
| LVIS [21] | 1,203 | 100,170 | 19,822 | $AP_r$, $AP_c$, $AP_f$, AP |
| COCO [34] | 80 | 118K | 5K | AP, $AP_{50}$, $AP_{75}$, $AP_s$, $AP_m$, $AP_l$ |
| Object365 [48] | 365 | 600K | 38K | AP, $AP_{50}$, $AP_{75}$, $AP_s$, $AP_m$, $AP_l$ |

pre-trained on the source dataset, which includes standard LVIS and ImageNet-21K (overlapping with LVIS). It is then directly used for inference on two target datasets: COCO [34] and Object365 [48]. We use AP, $AP_{50}$, $AP_{75}$, $AP_s$, $AP_m$, and $AP_l$ the evaluation metrics.

## D   Qualitative Results for Semantic Segmentation and Object Detection

In this section, we provide more qualitative results of our POMP for semantic segmentation and object detection. Figure 7 shows another three cases on open-vocabulary COCO-Stuff segmentation. POMP demonstrates a stronger ability than ZSSeg in the recognition of background classes. In case (1), POMP correctly identified the *dirt* and *plant-other* in the scene, instead of marking all these areas as *grass*. In case (2) and (3), POMP recognizes the classes of *clouds* and *tree*, respectively, while ZSSeg misclassifies them as *sky-other* and *bush*. However, POMP misses some objects of *sheep* located at the edge in case (2) and neglects the object of *branch* in case (3), indicating it still has insufficient recognition of small objects. For object detection, Figure 8 illustrates qualitative results on LVIS images. Base and novel categories are shown in purple and green, respectively. POMP identifies regions from the novel class without using the corresponding 1.2K detection annotations, demonstrating its generalization in the wild.

## E   More Ablation Study

### E.1   Ablation on Proposal Distribution

As introduced in § 3.2, we also investigate other types of proposal distribution for local contrast and negative class sampling. The first is the frequency distribution $Q^{(f)}$, which samples the negative class $i$ based on the number of training samples belonging to this class. Note that the original ImageNet-21K is class-imbalanced, i.e., the number of training samples belonging to common classes is larger than those belonging to rare classes, which can roughly reflect the long-tail distribution of object categories in nature. The frequency distribution will allow for more sampling of common classes while suppressing the exposure of rare classes in prompt tuning. Let $M_i$ be the number of

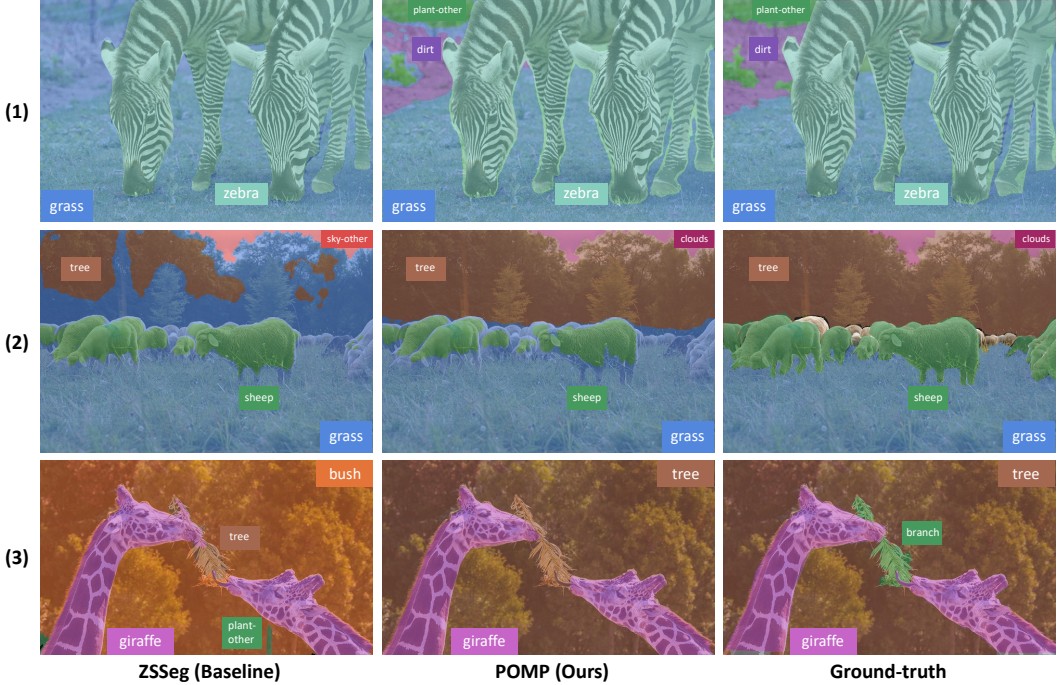

| ZSSeg (Baseline) | POMP (Ours) | Ground-truth |
| --- | --- | --- |

Figure 7: More qualitative results on open-vocabulary COCO-Stuff segmentation.

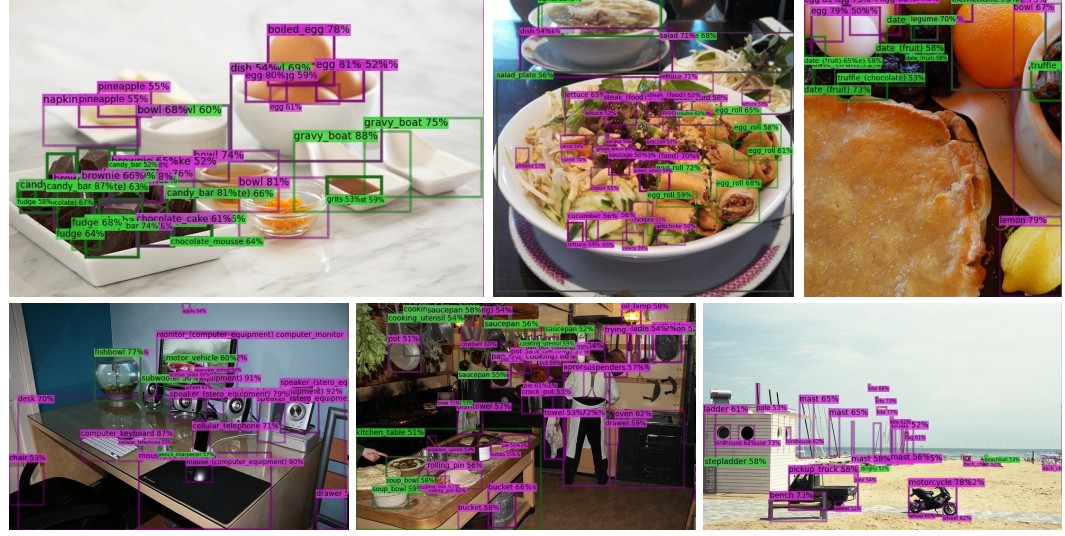

Figure 8: Qualitative results on LVIS images. Base and novel categories are shown in **purple** and **green** colors respectively. We use a score threshold of $0.5$ and show the most confident class for each box.

Table 12: Ablation on the sampling distribution in POMP based on CLIP (ViT/B-16) backbone.

| Method | ImageNet-21K | Cross-dataset (10 Avg.) | Cross-domain (4 Avg.) |
|---|---|---|---|
| POMP (uniform distribution) | 25.3 | 67.0 | 60.8 |
| POMP (frequency distribution) | 24.9 **(-0.4)** | 66.2 **(-0.8)** | 60.1 **(-0.7)** |
| POMP (similarity distribution) | 23.6 **(-1.7)** | 64.2 **(-2.8)** | 59.2 **(-1.6)** |

training samples belonging to the negative class $i$, the frequency distribution is defined as:

$$Q_i^{(f)} = \frac{M_i}{\sum_{j=1}^{N} M_j}.$$ (8)

The second is the similarity distribution $Q^{(s)}$, which aims to sample more hard negative classes. Hard negative classes are those that have a higher similarity between their features and the features of the input images, and are more likely to be confused with the positive class. Accordingly, in the similarity distribution, the likelihood of a negative class being sampled increases as the similarity between its feature and the image feature increases. To achieve this, we pre-encode features of all classes represented by a hand-crafted prompt (i.e., "a photo of a [CLASSNAME]"). The feature of class $i$ is denoted as $\mathbf{w}_i$. The likelihood of sampling a negative class is determined by the similarity between the class feature $\mathbf{w}_i$ and the image feature $\mathbf{x}$:

$$Q_i^{(s)}(\mathbf{x}) = \frac{\exp(\mathbf{x}^\top \mathbf{w}_i/\tau)}{\sum_{j=1}^{N} \exp(\mathbf{x}^\top \mathbf{w}_j/\tau)}.$$ (9)

Table 12 illustrates the performance of different proposal distributions. Compared to the uniform distribution, using the frequency distribution for sampling leads to degraded performance, particularly in cross-dataset and cross-domain settings, due to reduced sampling of rare categories. This highlights the importance of a large number of long-tail categories in the ImageNet-21K dataset for the generalization of the soft prompt. Additionally, the performance of the similarity distribution is also not as strong as that of the uniform distribution. The reason for this may be that as the soft prompt evolves, the features of hard negative classes change. However, the negative features used in (9) are obtained from the hard prompt, creating a fixed proposal distribution that is unable to adapt to these changes, potentially causing the soft prompt to converge to a local optimum. In contrast, POMP with the simple uniform distribution considers both common and rare classes, as well as easy and difficult classes, leading to the best performance for both the soft prompt and class features.

### E.2 Ablation on #shot and Prompt Length

We further conduct ablation on the number of pre-training instances per class (#shot) and the prompt length to analyze their influence on the generalization ability of POMP. The left panel in Figure 9 illustrates the results of #shot. The green curve represents the average accuracy of 10 datasets under the cross-dataset evaluation, while the purple curve represents the avergaed accuracy of 4 datasets under the cross-domain evaluation. Overall, the performance of POMP improves as #shot increases. We find that POMP can achieve decent cross-dataset and cross-domain accuracy even with #shot=1. This is due to the huge number of classes in ImageNet-21K. Even if there are only one instance per class, the overall amount of data (21K instances for 21K classes) is enough for training a soft prompt with only 0.012 M learnable parameters.

The right panel in the figure shows the results of the prompt length. The soft prompt of length 16 achieves 65.0% accuracy across datasets, which is lower than the soft prompt of length 4 with 67.0% cross-dataset accuracy. It indicates that the prompt with too large lengths impairs its generalization, which consistent with the findings from previous work [66, 29].

### E.3 Ablation on Prompt Types for Semantic Segmentation

We perform an ablation study on prompt types for cross-dataset semantic segmentation to further demonstrate the superior generalization ability of our prompt on downstream tasks. Specifically, we

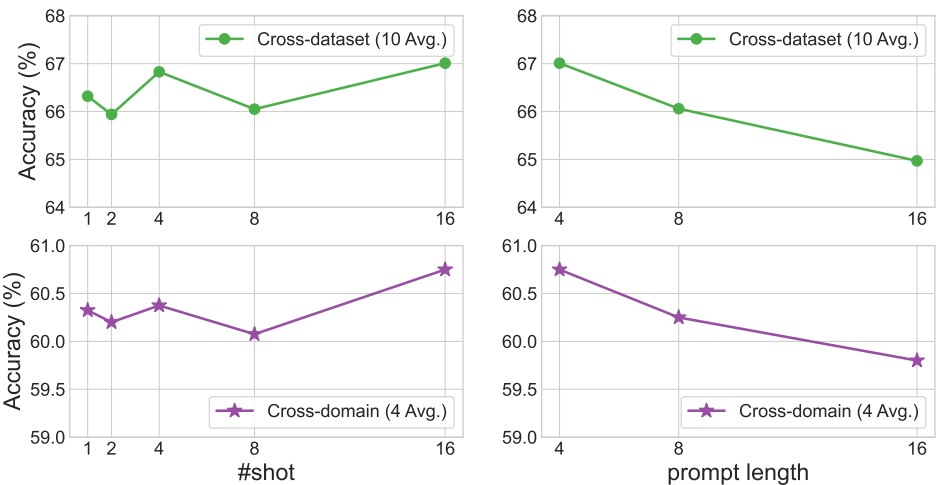

Figure 9: Ablation study on #shot and prompt length. When varying #shot, the prompt length is 4, and when varying the prompt length, #shot is 16.

Table 13: Cross-dataset evaluation for semantic segmentation. All methods share the same visual backbone with ZSSeg, but use different prompts.

| Method | Source Dataset: Standard COCO Stuff | | | | Target Dataset: ADE20K | | | | Target Dataset: PASCAL Context | | | |
|---|---|---|---|---|---|---|---|---|---|---|---|---|
| | mIoU | fwIoU | mACC | pACC | mIoU | fwIoU | mACC | pACC | mIoU | fwIoU | mACC | pACC |
| ZSSeg (single prompt) | 40.5 | 47.8 | 53.5 | 61.7 | 17.8 | 44.0 | 31.0 | 52.9 | 51.8 | 64.6 | 69.9 | 74.3 |
| ZSSeg (ImageNet prompts) | 40.9 | 48.4 | 54.7 | 62.3 | 17.7 | 46.5 | 31.8 | 57.1 | 52.0 | 64.7 | 70.3 | 75.4 |
| ZSSeg (ViLD prompts) | 40.9 | 48.6 | 54.2 | 62.3 | 20.2 | 49.1 | 33.4 | 60.7 | 51.8 | 63.8 | 69.6 | 73.8 |
| ZSSeg (POMP prompt, ours) | **41.2** | **49.0** | **54.7** | **62.6** | **20.6** | **49.3** | **35.0** | **61.7** | **52.4** | **65.3** | **70.6** | **76.4** |

take ZSSeg as the backbone and evaluate the performance of four types of prompts, as described in Appendix B. As shown in Table 13, ZSSeg with our POMP prompt achieves the highest performance on the three datasets. It is noteworthy that, despite using 80 hard prompts for *ImageNet prompts* and 14 for *ViLD prompts* for prompt ensemble, their performance was consistently worse than our POMP with just one soft prompt, highlighting the effectiveness of our method.

