# OpenReview forum: "Prompt Pre-Training with Twenty-Thousand Classes for Open-Vocabulary Visual Recognition"
_NeurIPS.cc/2023/Conference — NeurIPS 2023 poster_

### Official Review · Reviewer_xVVp · 2023-07-03

**Soundness:** 3 good
**Presentation:** 3 good
**Contribution:** 3 good
**Rating:** 6
**Confidence:** 4

**Summary:**

This paper proposed a vision-language pre-training method called POMP. It aims to solve the GPU memory problem of the existing vision-language pre-training method like CoOp when the class number is extremely huge. POMP is composed of local contrast and local correction strategies. Local contrast is to sample a few classes during training to save the GPU memory, while local correction is to alleviate the sampling bias problem. The authors provide very comprehensive experiments to show the effectiveness of the proposed pre-training method.

**Strengths:**

1. The proposed local contrast strategy is reasonable, easy to implement, and effective when the class number is huge.
2. The experiments are very comprehensive, and the code is also provided. So this work could provide valuable contributions to the open-source community.

**Weaknesses:**

The proposed local correction strategy is more like a general approach in the contrastive learning method. It encourages the positive samples to be much closer and pushes negative samples away for a more stringent decision boundary. So the reason why it works is that the features are more uniformly distributed in the feature space, which is supported by Fig. 5. Therefore, I do not think that the authors' claim that this strategy solves the sampling bias problem is proper, since the sampling bias still exists. Therefore, I encourage the authors to rethink the motivation and reason of the contrast correction strategy.

**Questions:**

None.

**Limitations:**

None.

---

> ### Author Rebuttal · Authors · 2023-08-10
>
> Thanks for your positive assessment and insightful comments.
>
> >**Q1: The claim that the local correction strategy solves the sampling bias problem is questionable.**
>
> A1: Thanks for your feedback. We agree that sampling bias still exists with our approach. However, our intention is to convey that the local correction helps mitigate negative effects stemming from sampling bias. As we describe in Section 3.2.2, class sampling causes the absence of some negative classes, which further leads to a more loose decision boundary and less-uniform class feature distribution.
>
> To alleviate this problem, our local correction strategy adds a margin term to create space for unsampled classes in the feature space. This encourages more discriminative representations even without observing all negative classes, which reduces the uniform loss from -1.05 to -1.37 and improves accuracy from 65.8 to 67.0 on ImageNet-21k (Fig.5). We will refine our statement of the local correction strategy in the final version.

---

> > ### Comment · Reviewer_xVVp · 2023-08-14
> > **Response to the rebuttal**
> >
> > Thanks for the clarification. My problem is solved and I keep my rating.

---

### Official Review · Reviewer_Mx1X · 2023-07-05

**Soundness:** 2 fair
**Presentation:** 2 fair
**Contribution:** 1 poor
**Rating:** 5
**Confidence:** 5

**Summary:**

* This paper present a prompt pre-training method for vision-language models named as POMP, which can be transfer to visual recognition tasks including image classification, semantic segmentation, and object detection.

* POMP follow the prompt tuning setting of CoOp. The authors claims that pre-training prompts on ImageNet-21K requires over 300 GB GPU memory with traditional methods like CoOp. To make it work and more efficient on ImageNet-21K, POMP propose a positive and negative class sampling to reduce gpu memory cost.

* POMP is prompted learning on ImageNet-21K dataset, also is also a task-agnostic prompt pre-training method, which can be directly used in downstream methods.

**Strengths:**

* The paper is well organized, the framework is simple.

* The authors have conducted a lot of experiments on downstream tasks.

* The design of local contrast and local correction is memory efficient for prompt tuning.

**Weaknesses:**

* The peformance of open vocabulary object detection & instance is really not high compared with detic(24.9 -> 25.2), even though POMP is prompt tuning on ImageNet21K dataset. The observation can be also validated in Table 6(Cross-dataset evaluation for object detection), For me, it seems that POMP don't work on detection benchmark.

* On detection experiments, POMP is based on detic, however, detic and POMP is not strictly open vocabulary method,  which means that the novel class names is known in training phase. For standard  open vocabulary tasks, the novel class can not be seem or trained, so the novel class can not be prompt tuned in this setting, How does POMP cope with this problem ?

* For me the novelty and the technical contribution of this paper is limited.

**Questions:**

please refer to weakness

---

> ### Author Rebuttal · Authors · 2023-08-10
>
> Thanks for your insightful comments. We humbly think that some concerns are caused by misunderstandings, which we will explain in detail below. We hope that our response can clarify the misunderstandings so that you can consider our work more favorably.
>
> >**Q1: The performance gain on the object detection tasks is limited.**
>
> A1: We note that the +0.3 mAPr gain on open-vocabulary LVIS is meaningful since Detic is a very strong baseline which uses extra ImageNet-21K data for detector pre-training. More importantly, on cross-dataset detection (Table 6), our POMP shows sizable gains of +1.9 AP50 on COCO and +0.8 AP50 on Object365.
>
> We would like to reiterate our main contribution that our method enables large-scale prompt learning for the first time, **rather than optimize detection scores**. We pre-train a universal prompt on ImageNet-21K that transfers broadly, achieving SOTA on 21 downstream tasks in zero-shot settings: the consistent gains across classification, detection, and segmentation tasks have offered strong evidence of its effectiveness.
>
> >**Q2: POMP takes Detic as backbone model for detection experiments. However, Detic is not a strictly open-vocabulary method for LVIS, how does POMP cope with this problem?**
>
> A2: Thanks for your feedback. We want to clarify our rationale for choosing the Detic backbone and experimental setup, and demonstrate that our improvement is consistent across various settings.
> 1. For the open-vocabulary LVIS task, Detic utilizes both box-supervised data from LVIS-base (866 classes) as well as image-supervised data from ImageNet-21K that overlaps with LVIS (997 classes, 277 of which are novel classes). This allows Detic to demonstrate transfer not only from base to novel classes, but also from image-level to box-level recognition. Since Detic is the closest existing method to ours that leverages ImageNet-21K, we chose it as a strong baseline and followed its setup for fair comparison.
> 2. To evaluate Detic and POMP more strictly on open-vocabulary LVIS, we conduct additional experiments training only on LVIS-base 866 classes using box-supervised data alone (denoted as **LVIS-base**. In contrast, Detic’s original setting is denoted as **LVIS-base & IN-L**). The results are as follow:
> | Method | Source training data | mAPr |
> |:-------:|:---------------:|:-----:|
> | Detic | LVIS-base | 16.4 |
> | POMP | LVIS-base | **17.4** |
> | Detic | LVIS-base & IN-L | 24.9 |
> | POMP | LVIS-base & IN-L | **25.2** |
>
> As shown, our method still shows consistent improvements over Detic, demonstrating the general efficacy of our approach for open-vocabulary detection. We would be happy to include these additional experiments in the final paper. Please let us know if you would like any clarification or have additional suggestions.
>
> 3. Additionally, we would like to highlight that our **cross-dataset experiments** (Table 6), training on LVIS-full and evaluating on COCO and Object365, **follow a strictly open-vocabulary protocol**.  As we reponsed in A1, our POMP achieves SOTA performance in these experiments as well, with gains of +1.9 AP50 on COCO and +0.8 AP50 on Object365 compared to prior art. In conclusion, we have provided strong evidence through additional experiments and analyses that our method advances the state-of-the-art in open-vocabulary detection across multiple datasets and protocols.
>
> >**Q3: The novelty and technical contribution are limited.**
>
> A3: We appreciate your feedback and want to clarify that our work makes several key innovations. First, our proposed methods of local contrast and local correction are novel techniques for efficient prompt learning at scale. More importantly, it is the combination and application of these techniques that enables prompt learning on large-scale visual concepts (e.g. ImageNet-21k) for the first time. This is a major contribution given that previous prompt tuning methods cannot feasibly scale to such a large number of concepts due to their computational complexity.
>
> By pre-training a universal prompt on ImageNet-21K, we obtain a prompt representation that transfers broadly and achieves SOTA results on 21 downstream tasks spanning classification, detection, and segmentation. We hope that our responses can clarify the misunderstandings and you will consider increasing your scores after seeing our responses. We'd appreciate any references or literature you can suggest, as we aim to cite them in our work.

---

> > ### Comment · Reviewer_Mx1X · 2023-08-21
> >
> > I agree that POMP enables large-scale prompt learning for the first time,  i will rise my rating.

---

> ### Author Response · Authors · 2023-08-19
> **Kind Reminder on Your Questions and Our Response**
>
> Dear Reviewer,
>
> Thanks again for your efforts in reviewing our paper.
>
> We provided our response one week ago. Does it address your questions? We are more than happy to answer any further questions.
>
> Thanks!

---

> ### Comment · Area_Chair_NurA · 2023-08-20
> **Reminder from AC**
>
> Dear Reviewer
>
> Could you read through the rebuttal and check if you have more questions / concerns ?
>
> Best,
> AC

---

### Official Review · Reviewer_i3py · 2023-07-07

**Soundness:** 3 good
**Presentation:** 3 good
**Contribution:** 3 good
**Rating:** 7
**Confidence:** 4

**Summary:**

This paper proposes POMP, a prompt pre-training method for pre-trained vision-language models like CLIP. While existing prompt-tuning approaches usually fine-tune the soft prompts on a specific downstream dataset with a limited number of classes, the proposed POMP conducts prompt "pre-training" on a large dataset (i.e., ImageNet-21K) with much more classes. POMP consists of two strategies: local contrast and local correction -- the former makes the memory footprint affordable with common GPUs, while the latter reduces the bias caused by the class sampling. Experiments show that POMP outperforms previous prompt-tuning methods across various visual recognition tasks and datasets.

**Strengths:**

- Extending prompt-tuning of vision-language models from task-specific datasets to a large "pre-training" dataset (e.g., the ImageNet-21K in this paper) is a promising direction, as existing prompt-tuning methods demonstrate poor cross-dataset generalization ability and always require per-task training.

- The proposed method is simple and has natural intuitions. The local contrast strategy makes the prompt tuning feasible with a large number of classes, but the aggressive class sampling introduces potential bias. Thus, the local correction is further proposed to alleviate the negative impact of the sampling bias. The definition of the adaptive margin m is clever and provides good insights.

- The experiments of this paper have wide coverage, which includes cross-dataset and cross-domain image classification, open-vocabulary object detection, and open-vocabulary semantic segmentation. The results can provide good references for future works in prompt tuning. Furthermore, the proposed POMP shows superior performance across various tasks and datasets, establishing a new state of the art for prompt tuning.

- The code of the proposed method and all baselines are included in the supplementary material.


**Weaknesses:**

I don't find significant flaws in this paper. There are some minor suggestions:

(1). Fig.1 looks a bit visually exaggerated: POMP only marginally improves the previous SOTA on some tasks and datasets, while the non-uniform scale in Fig.1 has a potentially misleading effect.

(2). Investigating the embedding space of POMP from the perspective of alignment and uniformity is interesting, but it is hard for me to parse "Class features of POMP have better alignment with centroids of the corresponding images, and are distributed with better uniformity" from Fig.6. Maybe it would be better to also add the concrete values of the alignment and uniformity loss.

**Questions:**

This paper makes solid contributions to the prompt tuning of pre-trained vision-language models. The proposed method is simple and well-motivated, and shows superior performance on a wide range of visual recognition tasks and datasets. The presentation of the paper is clear, with easy-to-understand figures and equations. Some minor suggestions are listed in the weaknesses section.

**Limitations:**

The limitations are discussed in the appendix of the paper. However, there is no discussions about the potential social impact, although the corresponding entry is checked in the checklist.

---

> ### Author Rebuttal · Authors · 2023-08-10
>
> Thanks for your positive assessment and constructive feedback.
>
> >**Q1: The non-uniform scale in Fig.1 has a potentially misleading effect.**
>
> A1: Thanks. Considering the tasks covered in the radar chart (Fig.1) have different difficulties and use different metrics (e.g., acc for classification, hIoU for segmentation), we follow previous work [1,2] to use independent scale for each task. To reduce potential misleading, we have updated the figure to use a consistent scale of 0.46 for all tasks except “Semantic Segmentation (Open-vocab PASCAL VOC)”. Given the large performance gain (+6.9) and range for this task, we have removed it from the figure to avoid compressing the other tasks. Please refer to our attached pdf.
>
> [1] CoCa: Contrastive Captioners are Image-Text Foundation Models. Yu et al. TMLR 2022.
>
> [2] Image as a Foreign Language: BEiT Pretraining for All Vision and Vision-Language Tasks. Wang et al. CVPR 2023.
>
> > **Q2: Avoid potential confusion in Fig. 6.**
>
> A2: Thanks for your suggestions. In Fig.6, our intention is to convey that:
> 1. Compared to light dashed lines (CoOp), the end points of solid lines (POMP) is closer to centroids of the points (image features), indicating that “class features of POMP have better alignment with centroids of the corresponding images''.
> 2. Compared to light dash-dot lines (CoOp), the angle between solid lines (POMP) is larger, indicating that “class features of POMP are distributed with better uniformity”.
>
> As you said, adding concrete values of the alignment and uniformity loss is a good solution. For Aircraft, POMP reduces the alignment loss $\ell_{\text{align}}$ from 1.39 to 1.36 and the uniform loss $\ell_{\text{uniform}}$ from -0.66 to -0.81. For UCF101, POMP reduces $\ell_{\text{align}}$ from 1.41 to 1.36 and $\ell_{\text{uniform}}$ from -0.95 to -1.23. We have updated the figure in our final version, please refer to our attached pdf.

---

> > ### Comment · Reviewer_i3py · 2023-08-20
> >
> > Thanks for the updated figures, I have no further questions and will keep my initial rating.

---

> ### Comment · Area_Chair_NurA · 2023-08-20
>
> Dear Reviewer
>
> Could you read through the rebuttal and check if you have more questions / concerns ?
>
> Best,
> AC

---

### Official Review · Reviewer_9929 · 2023-07-08

**Soundness:** 3 good
**Presentation:** 3 good
**Contribution:** 3 good
**Rating:** 5
**Confidence:** 4

**Summary:**

This paper introduce POMP, a prompt pre-training method for vision-language models. POMP learns a universal soft prompt that can express a large number of visual concepts and transfer to various visual recognition tasks in a zero-shot manner. POMP uses local contrast and local correction strategies to reduce the training cost and improve the generalization of the prompt. The paper shows that POMP outperforms previous state-of-the-art methods on several datasets and tasks, such as image classification, semantic segmentation, and object detection. The paper also analyzes the feature space of POMP and demonstrates its alignment and uniformity properties.


**Strengths:**

1. It proposes a novel and efficient method to pre-train a soft prompt on a large-scale dataset with over twenty-thousand classes, which enables the prompt to capture rich semantic information for visual recognition.
2. It introduces local contrast and local correction techniques to reduce the computational and memory overhead of prompt tuning
3. It achieves state-of-the-art performance on 21 datasets for various vision tasks, such as image classification, semantic segmentation, and object detection.

**Weaknesses:**

1. The training process for Stage 1 is similar to that of CLIP; however, CLIP utilizes a significantly larger pre-training dataset compared to ImageNet 22k. Therefore, it is important to clarify why the CLIP model cannot directly excel at these downstream tasks and why tuning it on ImageNet 22k could yield better results. Currently, it is unclear whether the performance improvement is primarily attributed to ImageNet 22k or the proposed prompt pre-training methods.
2. Figure 2 may inadvertently create misconceptions by suggesting that training solely on ImageNet 22k enables zero-shot transfer to tasks such as object detection and segmentation. In reality, as explained in the supplementary material, after completing Stage 1 training, an additional round of pre-training on the source data specific to object detection and segmentation is necessary. To avoid confusion, it is essential to provide explicit details regarding this additional training process.


**Questions:**

See weaknesses.

**Limitations:**

See weaknesses.

---

> ### Author Rebuttal · Authors · 2023-08-10
>
> Thank you for your thoughtful feedback. We hope that our response can address your concerns and you can consider our work more favorably.
>
> >**Q1: POMP’s pre-training process for stage 1 is similar to that of CLIP, why can't CLIP directly excel at downstream tasks compared to POMP, and why does tuning on ImageNet-22K yield better results?**
>
> A1: While POMP and CLIP both pre-train on image-text pairs, **there is a key difference in the text formatting.** CLIP uses paired captions, while POMP uses a "soft-prompt + [CLASSNAME]" format. This leads to two issues for CLIP when transferring to downstream tasks:
> 1. **CLIP requires extensive prompt engineering**, crafting hard prompts like "a photo of a [CLASSNAME]" for synthesizing classification weights. But hard prompts are unstable, as addressed in previous studies [1], small wording tweaks can drastically impact performance. For example, for the Caltech101 dataset, changing the prompt from “a photo of a [CLASS]” to “a photo of [CLASSNAME]” causes more than 5% decrease in accuracy.
> 2. **Generic prompts of CLIP lack task-relevant context**. For fine-grained/long-tailed datasets, prompts without contextual clues achieve lower accuracy. For example, for the EuroSAT dataset, the prompt of “a photo of a [CLASSNAME]” achieves 13% less accuracy compared to “a satellite photo of [CLASSNAME]”.
>
> In contrast, by pre-training a universal soft prompt on ImageNet-22K, POMP adapts better to downstream tasks without prompt engineering. The pre-trained prompt encodes broad coverage of visual concepts, providing more expressive context (see Line 252-254). This allows POMP to outperform CLIP on downstream transfer without the need for extensive fine-tuning.
>
> [1] Learning to Prompt for Vision-Language Models. Zhou et al. IJCV 2022.
>
> >**Q2: Avoid potential confusion in Fig.2**
>
> A2: Thanks for your suggestion. As you note, for detection and segmentation tasks, the region proposal and mask proposal networks need to be pre-trained on detection and segmentation data, respectively, and the hard prompt (e.g., “a photo of a”) should be replaced with our POMP prompt. We have updated the caption of Fig.2 with these notes in our final revision (see our attached pdf).

---

> > ### Comment · Reviewer_9929 · 2023-08-20
> >
> > The response has addressed my concerns. I keep the positive rating.

---

### Author Rebuttal · Authors · 2023-08-10

We want to thank all the reviewers for the positive assessment and insightful comments, which helped improve the quality of our work. We have revised our paper accordingly and provided individual responses to each reviewer. Please find attached a PDF outlining the main changes:

1. Updated Fig.1 with a consistent scale.
2. Clarified in Fig. 2 caption that the region/mask proposal networks require pre-training on detection/segmentation source data, as suggested.
3. Added concrete values of the alignment and uniformity loss in Fig. 6.

We hope that our responses can clarify the misunderstandings and help the reviewers consider our work more favorably.

---

### Author Response · Authors · 2023-08-21
**General Reply to Area Chairs and Reviewers**

We greatly appreciate the effort of the area chairs in coordinating the discussion process. We are also very grateful to the reviewers for thoroughly reading our rebuttal and appreciating the value of our work. If there are any remaining questions or comments, please don't hesitate to let us know.

Thanks, Authors

---

### Decision · Program_Chairs · 2023-09-21

**Decision:**

Accept (poster)

**Comment:**

Reviewers have come to a consensus on accepting this paper, the authors are expected to incorporate the suggestions from reviewers in the final camera ready version.